# Leveraging machine learning for quantitative precipitation estimation from Fengyun-4 geostationary observations and ground meteorological measurements

Xinyan Li[1], Yuanjian Yang[1], Jiaqin Mi[1], Xueyan Bi[2], You Zhao[1], Zehao Huang[1], Chao Liu[1], Lian Zong[1], and Wanju Li[2]

1School of Atmospheric Physics, Nanjing University of Information Science and Technology, Nanjing, 210044, China
2Institute of Tropical and Marine Meteorology, China Meteorological Administration, Guangzhou, 510080, China

*Correspondence to*: Yuanjian Yang (yyj1985@nuist.edu.cn)

**Abstract** Deriving large-scale and high-quality precipitation products from satellite remote sensing spectral data is always challenging in quantitative precipitation estimation (QPE), and limited studies have been conducted even using the China's latest Fengyun-4A (FY-4A) geostationary satellite. Taking three rainstorm events over South China as examples, a machine-learned regression model was established using the Random Forest (RF) method to derive QPE from FY-4A observations, in conjunction with cloud parameters and physical quantities. The cross validation results indicate that both daytime (DQPE) and nighttime (NQPE) RF algorithms performed well in estimating QPE, with the bias score, correlation coefficient and root-mean-square error of DQPE (NQPE) of 2.17 (2.42), 0.79 (0.83) and 1.77mm/h (2.31mm/h), respectively. Overall, the algorithm has a high accuracy in estimating precipitation under heavy rain level or below. Nevertheless, the positive bias still implies an overestimation of precipitation by the QPE algorithm, in addition to certain misjudgements from non-precipitation pixels to precipitation events. Also, the QPE algorithm tends to underestimate the precipitation at the rainstorm or even above levels. Compared to single-sensor algorithm, the developed QPE algorithm can better capture the spatial distribution of land-surface precipitation, especially the centre of strong precipitation. Marginal difference between the data accuracy over sites in urban and rural areas indicate that the model performs well over space and has no evident dependence on landscape. In general, our proposed FY-4A QPE algorithm has advantages for quantitative estimation of summer precipitation over East Asia.

# 1 Introduction

Precipitation is an important element of weather and climate systems, as well as the global cycling of water and energy (Hobbs, 1989; Fu et al., 2017; Yang et al., 2021). Accurate precipitation observations are important to industrial and agricultural production, water use, and flood and drought monitoring (Behrangi et al., 2014; Gan et al., 2016; Lolli et al., 2020). Traditional ground-station observations of precipitation possess extremely high measurement accuracy on the point scale, but they cannot accurately reflect the precipitation on the surface scale owing to the sparse distribution and network density of stations (Li et al., 2013; Liu et al., 2013). Ground-based radar observations can give the spatial and temporal distribution of precipitation within a 300-km radius range, but their spatial coverage cannot be scaled up to the global scale (Lee et al., 2015). With the rapid development of remote sensing, meteorological satellites have become the only viable way to observe precipitation globally at both high spatial and temporal resolution (Tang et al., 2016; Hou et al., 2014). However, large-scale and high-quality precipitation products derived from satellite remote sensing spectral data have always been a challenging issue in satellite quantitative precipitation estimation (QPE) (Lensky and Rosenfeld, 1995; Min et al., 2019).

With the constant improvement of meteorological satellites, satellite-based QPE technology has developed greatly. QPE satellite spectrum precipitation retrieval algorithms can be divided into visible/infrared (VIS/IR), microwave, and multi-combined spectral signals (Kidd, 2010; Levizzani et al., 2007). VIS/IR precipitation retrieval algorithms mainly include the Geostationary Operational Environmental Satellite (GOES) Precipitation Index algorithm (Arkin and Meisner, 2009), the GOES Multispectral Precipitation Algorithm (Ba et al., 2001), the Griffith–Woodley algorithm (Griffith et al., 1978), and the Climate Estimation Centre Merged Analysis of Precipitation algorithm (Xie and Arkin, 2001). Rosenfeld and Gutman (Rosenfeld and Gutman, 1994) explored the relationship between the effective radius of cloud retrieved by NOAA satellites and precipitation, and proposed that an effective radius greater than 14 μm should be the threshold for precipitation in the cloud. Previous studies have shown that different cloud microphysical parameters are closely related to the ground-level precipitation intensity, such as the substantially positive correlation between cloud optical thickness/cloud liquid water content/cloud effective radius and the surface rain rate, while there is basically a negative correlation between the cloud-top temperature and surface rain rate (Fu, 2014; Nauss et al., 2008; Rosenfeld and Gutman, 1994; Rosenfeld et al., 2012; Yang et al., 2018). Microwave precipitation retrieval algorithms include passive microwave (PMW) precipitation retrieval methods such as the Ferraro algorithm (Ferraro and Ralph, 1997), Goda profile algorithm (Kummerow et al., 2001), and the Passive Microwave Neural Network Precipitation Retrieval approach for The EUMETSAT Satellite Application Facility on Support to Operational Hydrology and Water Management (H-SAF) (Mugnai et al., 2013), as well as active precipitation retrieval methods based on the Precipitation Radar (PR) carried onboard the Tropical Precipitation Measuring Mission satellite (Iguchi et al., 2000) and The Global Precipitation Measurement (GPM) Core Observatory spacecraft (Sharifi et al., 2016; Tan and Duan, 2017). Based on the higher temporal sampling frequency of geostationary satellites, VIS/IR algorithms are suitable for retrieving continuous precipitation (Kidd, 2010), while PMW algorithms are better for retrieving instantaneous precipitation with higher accuracy (Ebert and Manton, 1996; Bauer et al., 1995), although PR also has the disadvantage of a limited observation range and uncertain parameters (Iguchi et al., 2009). Therefore, the development of multi-spectral joint

precipitation inversion algorithms can make up for the shortcomings of single-sensor algorithms (Michaelides et al., 2009; Holl et al., 2010).

Because precipitation is a highly complex process, however, there is a nonlinear relationship between the surface precipitation intensity and cloud-top optical physical variables, resulting in certain limitations in the precipitation-estimation equation constructed with statistical methods (Atkinson and Tatnall, 1997). Machine learning is widely used in satellite QPE (Kühnlein et al., 2014; Min et al., 2019; Chen et al., 2019; Zhang et al., 2019; Sanò et al., 2015), and the Random Forest (RF) model is a modern machine-learning technique for classification and regression, as well as a combined self-learning technique, which can easily capture the complex nonlinear relationship between observational and meteorological-environmental elements (Breiman, 2001; Bai et al., 2019a, 2019b). It has been widely applied to QPE. For instance, Kühnlein et al. (2014)divided data from the Spinning Enhanced Visible and Infrared Imager carried onboard the Meteosat Second Generation satellite into day, dusk and nighttime to establish an RF model and carry out QPE research, the results of which demonstrated a good effect on the estimation of rain area and convective precipitation. Min et al. (2019) used Himawari-8 real-time multi-band infrared brightness temperature and the Global Precipitation Measurement product to establish a QPE method based on the RF model, from which it was found that the accuracy of distinguishing the precipitation area reached 0.87 and its average absolute error and mean square error were 0.51 mm/h and 2.0 mm/h, respectively. Thus, there is strong evidence that the RF model can be applied effectively in precipitation monitoring and forecasting. The Fengyun-4 satellite (FY-4A), launched in December 2016, is China's second-generation geostationary meteorological satellite, and carried onboard is the Advanced Geostationary Radiation Imager (AGRI) with 14 spectrum detection bands, covering the visible light, shortwave infrared, midwave infrared, and longwave infrared bands. Thus far, QPE-based research using FY-4A remains limited, especially in terms of the lack of an RF-based FY-4A QPE framework.

South China is one of the regions in the country with the longest rainy season, the most abundant precipitation, and frequent heavy rains. Affected by the westerly wind system and the East Asian subtropical monsoon, the period from April to June each year is the first rainy season (or the first flood season) in South China. Therefore, it is important to strengthen the study of precipitation estimation and monitoring methods in the first flood season in South China. In the present work, taking South China (109°–118°E, 20°–26°N) as the research area, a RF algorithm model for FY-4A QPE is proposed by using the spectral reflectance observations of FY-4A/AGRI, meteorological environmental physical quantities from the fifth major global reanalysis produced by the European Centre for Medium-Range Weather Forecasts (ERA5), and a precipitation dataset observed by a high-density automatic station network with hourly resolution. The aims of this study are to further improve the multi-spectral monitoring level of the FY-4A satellite and provide a scientific basis for improving the disaster prevention and mitigation capabilities of the FY-4A satellite.

## 2 Data and methods

### 2.1 Rainstorm cases

Rainstorms occurred frequently during the first flood season in South China in 2019, causing huge losses of life and property. We selected three rainstorms in South China during 2019 that had a long period of precipitation with a large coverage area, as follows: Case 1: April 11–12, 2019 (Beijing time; if not specified, Beijing time is used); Case 2: June 12–13, 2019; Case 3: June 23–24, 2019. In addition to 215 national operated meteorological stations, there are also 4,706 automatic stations over the study region, with a mean distance between them less than 10 km (Figure 1). Also, stations are deployed with higher density in the urban built-up area with relative to the rural area. For the high-density automatic station precipitation data in the range of 109°–118°E and 20°–26°N, after filtering and deleting the missing and misdetected data, the number of automatic stations for the three cases was 4263, 4610 and 4623, respectively. The distribution of high-density automatic stations throughout the country and in the research area is shown in Figure 1, and the spatial distributions of precipitation during the three South China rainstorms are shown in Figure 2. The red shading area in Figure 1 shows the build-up area in South China, which was extracted using nighttime light (NTL) data obtained by the Visible Infrared Imaging Radiometer Suite (VIIRS) on board the Suomi National Polar Orbiting Partnership (NPP) satellite. In Case 1 (Figure 2a), large-scale heavy precipitation mainly occurred in the southern coastal area of Guangdong Province. From north to south, there were three bands of precipitation extremes, and the accumulated precipitation gradually decreased from southwest to northeast. In total, 159 automatic stations recorded precipitation exceeding 100 mm within 48 h, and the maximum precipitation in Jiujiang Town, Foshan City (112.99°E, 22.83°N), on the order of 192 mm, reached rainstorm levels (The specific rainfall classification of was supplied in table S1 in the supplementary materials).

In Case 2 (Figure 2b), there was a belt of accumulated heavy precipitation in the northwest mountainous area, and a large area of heavy precipitation in the northeast of the central urban area. The 48-h automatic station precipitation amounts in these two concentrated heavy precipitation areas both exceeded 100 mm, which meant that the precipitation intensity met the heavy rain level (see table S1, so does the following description of precipitation level). The precipitation of 492 automatic stations exceeded 100 mm within 48 h, and the maximum precipitation was 318.7 mm in Fogang County, Qingyuan City (113.93°E, 23.91°N).

In Case 3 (Figure 2c), there were three heavy precipitation areas that met the heavy rain level. The heavy precipitation area on the west side extended from Yulin City to the southeast to the north of Maoming City. The distribution of high-density automatic stations in this area is sparse. The longitude and latitude of the maximum precipitation station were 111.04°E, 22.59°N, and the precipitation in 48 hours was 149.3mm. A central heavy precipitation area covered Guangzhou and its surrounding urban areas with high population density, where the distribution of automatic stations is extremely dense. The longitude and latitude of the maximum precipitation station were 113.57°E,23.28°N, and the precipitation in 48 hours was 220.5mm. On the east side, a strong precipitation centre formed in the southwest of Longyan City and connected with Meizhou City to the west. The underlying surface is mountainous, meaning that the distribution of automatic stations in this area is sparse. The longitude and latitude of the maximum precipitation station were 116.17°E,24.65°N, and the

precipitation in 48 hours was 239.4 mm. Three strong rainfall areas obviously met the heavy rain levels. The precipitation of 644 automatic stations exceeded 100 mm within 48 h, and the maximum precipitation was 239.4 mm in Haizhu District of Guangzhou (113.30°E, 23.10°N).

Among the three cases, Case 1 had a small distribution of heavy precipitation, and the accumulated precipitation was the smallest among the three cases. Case 2 had 29 stations with a 48-h accumulated precipitation exceeding 200 mm, and with more extreme heavy precipitation. Case 3 had the largest number of stations meeting the rainstorm level and had a wide distribution. The types of underlying surface covered by the precipitation were diverse, and the centre of the rainstorm was located in the central urban agglomeration and densely populated areas, which meant that the threat to human life and property was high, so Case 3 is more typical and representative to be as a study case with respect to other two cases. This paper therefore takes the heavy rain process of June 23–24 as an example to discuss the QPE method of FY-4A based on the RF model and physical quantities. Theestimation and validation results of the RF model for Case 1 and Case 2 are provided in the supporting information.

## 2.2 Data

The 14 bands of FY-4A/AGRI have different detection purposes and can identify different spectral characteristics of different surfaces, clouds or atmospheres (Table S2 in the supporting information). FY4A/AGRI takes about 15 minutes to perform a full-disk image observation and has a maximum spatial resolution of 500 m. FY4A/AGRI provides level-1 dataset with resolutions of 500 m, 1 km, 2 km and 4 km at nadir, and 4 km at nadir for level-2 dataset, which meet the requirements for the spatial and temporal resolution of satellite monitoring of rainstorms. In order to train the RF model, we used the FY-4A/AGRI full disk data with a temporal resolution of 1 h and spatial resolution of 4 km × 4 km during the study period, which contains 14 bands of radiation brightness temperature and reflectance information. At the same time, the combined channel information was constructed based on the level-1 data.

Due to the indirect link between the surface rain rate and the cloud-top brightness temperature (Boris et al., 2008), the inversion accuracy is limited. To ensure stability of training and estimation, as well as improve the estimation accuracy, according to existing research (Kühnlein et al., 2014; Min et al., 2019), four level-2 cloud parameter products [cloud-top temperature (CTT), cloud-top height (CTH), cloud type (CLT), and cloud phase (CLP)] observed in real-time from FY-4A were selected. For each cloud parameter product, the temporal resolution is 1 h and the spatial resolution 4 km × 4 km, which is consistent with the 1-level data. CTT and CTH are the cloud-top temperature and height information of cloud pixels obtained by inversion of AGRI infrared channel data, which can be used to determine the likelihood of cloud growth, extinction, and precipitation. CLT is four different cloud phases generated from AGRI infrared channel data—namely, warm liquid water (> 0℃), supercooled liquid water, mixed, and ice. CLP uses data from multiple infrared channels of AGRI to obtain six different cloud types through a series of spectral and spatial tests: water, supercooled water, mixed, thick ice, thin ice, and multi-layer ice. CLT and CLP are commonly used to detect and track changes in water vapor composition in clouds and extreme weather estimation to improve extreme weather warning capabilities.

In addition to the FY-4A/AGRI observation data, this paper also uses physical quantities from ERA5 to further improve the performance of the QPE algorithm. These data have a horizontal resolution of 0.25° × 0.25°, a vertical resolution of 37 layers, and a temporal resolution of up to 1 h. ERA5 is widely used in the study of weather and climate change. According to previous studies (Min et al., 2019; Kanamitsu and Masao, 1989), we introduce some ERA5 reanalysis data to further and better support QPE, including six physical weather indexes, which can effectively describe the atmospheric heat (K-Index), dynamics [convective available potential energy (CAPE); eastward turbulent surface stress (EWSS)], humidity [total column water vapour (TCWV); total column water (TCW)] and topographic features [anisotropy of sub-gridscale orography (ISOR)]. These indexes are closely associated with the initiation and development of clouds that produce rain (Zhang and Guang, 2003; Roman et al., 2016). In order to improve the generalization ability of the algorithm precipitation estimation and solve the difference of algorithm precipitation estimation in plain and mountain areas, this paper introduces the Digital Elevation Model (DEM) as one of the input variables. It is the digital expression of surface morphology and contains rich topographic and geomorphic information.

## 2.3 RF model design

A data-driven regression model was established between the observed precipitation and satellite data as well as cloud parameters using the RF method. The essence of the RF data estimation model is as follows:

1) The input variables to the RF model are shown in Table 1, including geographic information, channel information, combined channel information, cloud parameter products, and ERA5 data. A Daytime Quantitative Precipitation Estimate (DQPE) algorithm and a Nighttime Quantitative Precipitation Estimate (NQPE) algorithm were constructed separately,due to different input variables between daytime and nighttime. The DQPE algorithm is used to estimate the precipitation from 8:00 to 16:00, and the NQPE algorithm is used to estimate the remaining time periods. The visible light channel at nighttime cannot produce valid observational information, so the NQPE algorithm removes these variables. The CTT gradient ($CTT_G$) in the combined channel information is closely related to the rain rate, defined as follows Eq. (1):

$$CTT_G = \{[T(i+1,j) - T(i-1,j)]^2 + [T(i,j+1) - T(i,j-1)]^2\}^{\frac{1}{2}}, \tag{1}$$

where $T$ represents the spectral brightness temperature of 10.7 μm, and $i$ and $j$ represent the pixel position.

2) We selected the 1-h temporal resolution high-density automatic station geographic and precipitation information, satellite observation data and ERA5 reanalysis data to input into QPE algorithm for precipitation estimation. The spatial resolution of FY-4A/AGRI is 4 km × 4 km, the spatial resolution of ERA5 is 0.25° × 0.25°, the spatial resolution of DEM is 1 km × 1 km. Due to this difference in spatial resolution, the abovementioned data needed to be interpolated to construct a dataset that was synchronized in time and space. According to previous study (Liu et al., 2020), the differences between a diverse of interpolation methods are small for high-density automatic stations, with the effect of interpolation depending mainly on the station distribution rather than the interpolation method itself. In this paper, for matching input variables with precipitation data, we employed spline interpolation on the satellite data to match the in-

situ precipitation measurement, while used the averaged value of four nearest grids of ERA5 data and DEM data around each weather station to match the in-situ precipitation measurement/ satellite data at each pixel.

3) Ten indicators are defined to judge the accuracy of the QPE algorithm (Table 2). In order to quantitatively evaluate the classification results of precipitation and non-precipitation pixels, we introduce eight classical metrics: bias score (*Bias*, *Bias* = 1 unbiased, *Bias* < 1 underestimation, *Bias* > 1 overestimation), probability of detection (*POD*, optimal = 1), false alarm ratio (*FAR*, optimal = 0), accuracy (*ACC*, optimal = 1), Critical Success Index (*CSI*, optimal = 1), Heidke Skill Score (*HSS*, optimal = 1), Hanssen and Kuiper (*HK*, optimal = 1), and Equitable Threat Score (*ETS*, optimal = 1). Among them, when *POD* or *FAR* take the optimal value, the algorithm cannot be determined as the optimal estimation. When *ACC* or *CSI* take the optimal value, the algorithm can be determined as the optimal estimation. *HSS*, *HK* and *ETS* are all commonly used to evaluate the estimated ability of algorithms as skill scores. *HSS* compares the accuracy between the algorithm and a random estimation as reference by the accuracy score *ACC*, and *ETS* compares the accuracy between the algorithm and another random estimation as reference by the accuracy score *CSI*. When *HSS* > 0 or *ETS* > 0, the algorithm is skillful and its estimation ability is better than random estimation as reference. HK is defined as the difference between *POD* and probability of false detection (*POFD*). When *HK* > 0, the algorithm is skillful, and when *HK* = *HSS*, the algorithm is unbiased estimation. The other two indicators can be used to quantitatively evaluate the accuracy of precipitation estimation based on the QPE algorithm. They are correlation coefficient (*R*, optimal = 1), and root-mean-square error (*RMSE*, optimal = 0).

4) The establishment of the RF model needs to determine two important parameters—namely, the number of input variables of tree nodes, *mtry*, and the number of decision trees, *ntree*. Besides, the larger the *mtry*, the smaller the overfitting effect of the RF algorithm; while the larger the *ntree*, the smaller the difference between the submodels. The value of *mtry* should be smaller than the value of the input variable. In general, *mtry* values are 1, $k/2$, $\sqrt[2]{k}$ and $\log_2(k) + 1$ etc., where $k$ is the number of variables input into the model. We selected $k/2$ as the number of input variables of the tree node; that is, *mtry* = 17. The number of decision trees, *ntree*, is ideally classified when the value of *ntree* is between 500 and 800. We set it to 550.

5) Randomly create *mtry* pieces of variables for the binary tree on the node, and the choice of the binary tree variables still meets the principle of minimum node impurity. The Bootstrap self-help method is applied to randomly select *ntree* sample sets from the original dataset to form a decision tree of *ntree*, and the unsampled samples are used for the estimation of a single decision tree. Samples are classified or predicted according to the RF composed of *ntree* decision trees. The principle of classification is the voting method, and the principle of estimation is the simple average."

Finally, figure 3 summarizes the flowchart for the QPE algorithm using the RF model. According to previous studies (Yang et al., 2020; Zeng et al., 2020), a ten-fold cross-validation (10-fold CV) method was used to test the model estimation performance. The 10-fold CV method makes maximum use of the existing sample data and ensures that each sample is used as a training sample and a test sample respectively, effectively avoiding the result of over-fitting. The training set was input into the RF model, and the QPE algorithm with the highest estimation accuracy was constructed by performing 10-fold CV. The testing set was input into the RF model to obtain the precipitation estimation of each pixel and judge whether the pixel

was a precipitation pixel. For a pixel with precipitation intensity greater than 0.1 mm/h, it was judged as a precipitation pixel; otherwise, it was judged as a non-precipitation pixel.

## 3 Results

### 3.1 RF model evaluation

This paper uses 10-fold CV to evaluate the accuracy of precipitationestimation. Figure 4 compares the precipitation observations of the high-density automatic stations in the training set and the testing set with the precipitation estimation of the QPE algorithm in the 10-fold CV of the DQPE algorithm and the NQPE algorithm, in which the color bar represents the occurrence frequency on a log scale with an interval of 0.5 mm/h. In the training set, the *Bias*, *R* and *RMSE* of the DQPE (NQPE) algorithm are 1.75 (1.95), 0.97 (0.98) and 0.77mm/h (1.00mm/h), respectively. In the testing set, the *Bias*, *R* and 235   *RMSE* of the DQPE (NQPE) algorithm are 2.17 (2.42), 0.79 (0.83) and 1.77mm/h (2.31mm/h) respectively. For the heavy rain level and below precipitation observation samples, the large amount of precipitation observations and estimations are close to the 1:1 line. It reflects that the algorithm has a high quantitative estimation ability for precipitation of heavy rain level and below. *Bias* are all greater than 1, reflecting that the QPE algorithm tends to overestimation precipitation events. The feature of the algorithm overestimation precipitation events is mainly manifested in weak precipitation and non-240   precipitation pixels. For example, there are obviously a large number of observation sample points without precipitation in Figure 4, and their precipitation estimation is too high. When the precipitation observations reach the rainstorm level and above, the QPE algorithm tends to underestimate the precipitation. This kind of estimation error can be reduced by secondary training. In addition, it is found that there exists a notable difference in the performance between testing and training for the QPE algorithm. This gap is suspected to be caused by over-fitting that is possibly related to the high 245   complexity of the RF model (Lao et al., 2021). The effect of the QPE algorithm on Case 1 and Case 2 is shown in Figure S1 of the supplementary materials.

Table 3 shows the evaluation metrics in training set and testing set of DQPE (NQPE) algorithm. The *POD* of the QPE algorithm are 0.98 and above, and the *FAR* are around 0.5. For the precipitation pixels, the QPE algorithm can accurately identify them; but for the precipitation pixels estimated by the QPE algorithm, the probability that the pixel has precipitation 250   is about 50%. The *ACC* reach 0.6 and above, indicating that for more than 60% of the data samples, the QPE algorithm can accurately distinguish between precipitation and non-precipitation pixels. The *CSI* are 0.4 and above, which means that the accurately estimated precipitation pixel samples account for more than 40% of all observed and estimated precipitation pixel samples. *HK* reflects the difference between *POD* and probability of false detection (POFD). The *HK* of the QPE algorithm are around 0.5 and above, indicating that the proportions of correct estimations in all precipitation pixels are far greater than 255   the proportions of incorrect estimations in all non-precipitation pixels. *HSS* (*ETS*) reach 0.3 and above (0.2 and above), reflecting that the QPE algorithm is skillful and better than random estimation as reference ($A_{ref1}$ and $A_{ref2}$ in Table 2). The evaluation metrics of the QPE algorithm on Case 1 and Case 2 are shown in Table S2 of the supporting information.

**3.2 Application of the RF model to QPE**

Figure 5 shows the hourly precipitation distribution as estimated by the DQPE algorithm, and Figure 6 shows the actual precipitation observations at high-density automatic stations in the daytime, with a temporal resolution of 1 h. Comparison of Figures 5 and 6 shows that the precipitation estimation of the DQPE algorithm is consistent with the actual precipitation observations at high-density automatic stations, and the DQPE algorithm can capture the precipitation range well. However, when the precipitation exceeds 20 mm/h, the algorithm obviously underestimates the precipitation. This means that the algorithm can only roughly estimate the location and range of extreme precipitation pixels, but cannot accurately and quantitatively estimate extreme precipitation. This is similar to the results of previous studies (Kühnlein et al., 2014; Min et al., 2019). At 11:00–16:00 on June 24, the precipitation centre gradually moved to the sea surface. Due to the lack of geographic information provided by the high-density automatic stations for training at the sea surface, the accuracy of the estimation is low. However, for land-surface precipitation, the size, location and coverage of the precipitation estimated by the DQPE algorithm is highly consistent with the actual precipitation observations. The precipitation estimation of Case 1 and Case 2 by the DQPE algorithm is shown in Figure S2 of the supporting information. The distribution of the measured precipitation in the daytime from the high-density automatic stations in Case 1 and Case 2 is shown in Figure S3 of the supporting information.

Figure 7 shows the hourly distribution of precipitation as estimated by the NQPE algorithm, and Figure 8 shows the actual precipitation observed at the high-density automatic stations at nighttime. Comparing Figures 7 and 8, a conclusion similar to that from the DQPE algorithm can be obtained. The precipitation estimation of Case 1 and Case 2 by the NQPE algorithm is shown in Figure S4 of te supporting information. The distribution of precipitation at nighttime observed by the high-density automatic stations in Case 1 and Case 2 is shown in Figure S5 of the supporting information. In general, the estimation ability of the QPE algorithm is strong over the land surface. This proves the applicability and feasibility of establishing an RF model and training the QPE algorithm based on the model variables in Table 1.

**3.3 Verification of the QPE**

In order to further analyse the factors that affect the estimation accuracy of the QPE algorithm, we selected three city stations as research targets: Guangzhou (113.30°E, 23.13°N), Shantou (116.50°E, 23.38°N), and Zhuhai (113.57°E, 22.28°N). At the same time, we selected three rural stations as research targets: Datian (117.93°E, 25.80°N), Lianshan (112.03°E, 24.63°N), and Jinxiu (109.99°E, 24.09°N). Figure 9 shows the actual precipitation observations of these six stations and the accumulated precipitation estimated by the QPE algorithm for 48 consecutive hours. Based on the precipitation observations and estimations at each time of 48 hours, we used *Bias*, *R* and *RMSE* to evaluate the ability of the algorithm for precipitation estimation. For the six research sites, *Bias* are all greater than 1, reflecting that the algorithm tends to overestimation precipitation events, which is consistent with the conclusion above. Based on *R* and *RMSE*, there is no obvious difference between city sites and rural sites. The time and size of the cumulative precipitation changes in the six research sites are almost the same, reflecting that the algorithm has a strong quantitative estimation ability and does not change with city and

rural areas. We can think that the precipitation estimation ability of the algorithm is less affected by the difference between city and rural areas.

Figure 10(a) presents the 48-h accumulated precipitation estimated by the QPE algorithm. Compared with Figure 2(c), all three heavy precipitation centres are estimated, and the area and intensity in the precipitation estimations and in the actual observations are basically the same. It shows that the algorithm has strong potential in accurately estimating the intensity and range of precipitation. Figures 10(b) and 10(c) respectively represent the actual precipitation frequency observed by the high-density automatic stations and that estimated by the QPE algorithm. The results indicate that the frequency of precipitation in the northeast of the study area is relatively greater, and vice versa in the southwest. The precipitation frequency estimated by the QPE algorithm is generally greater than observed. This is because there are more non-precipitation events for most stations and the algorithm often incorrectly judges non-precipitation areas as weak precipitation stations as *Bias* is greater than 1, resulting in a positive bias in the precipitation frequency estimated by the QPE algorithm at each station. The spatial distribution of accumulated precipitation in Case 1 and Case 2 is shown in Figure S6 of the supporting information.

Figure 11 shows the spatial distribution of evaluation indicators of the QPE algorithm for all stations. Except that the *POD* of almost all stations are close to 1 in Figure 11(a), the spatial distribution of the rest of the evaluation indicators has obvious correlation. In Figure 11(b) and Figure 11(i), the spatial distribution of *FAR* and *Bias* has a significant negative correlation with accumulated precipitation. *FAR* is often lower and *Bias* tends to 1 in areas with more accumulated precipitation, such as the three heavy precipitation centres in the precipitation process mentioned above. On the contrary, the *FAR* and *Bias* are higher in areas with less accumulated precipitation, such as the southwest coastal area. According to the *FAR* and *Bias* calculation formula (Table 2), this reduces the *FAR* and *Bias* by increasing the number of precipitation pixels detected by both stations and the QPE algorithm There is a *low-FAR* and *low-Bias* zone in the mountainous area in the northwest of the study area, which does not meet the above characteristics. Combined with the results shown in Figures 8(a) to 8(h), we can see that there has been weak precipitation in the area for at least eight consecutive hours. According to the *FAR* and *Bias* calculation formula (Table 2), this reduces the *FAR* by reducing the number of precipitation pixels that are not detected by the stations but are detected by the QPE algorithm. Therefore, *FAR* and *Bias* is negatively correlated with precipitation intensity and duration. When the precipitation intensity is greater and the duration is longer, the *FAR* of the QPE algorithm is lower, the *Bias* gets closer to 1, and the deviation that the QPE algorithm accurately distinguish precipitation and non-precipitation pixels is smaller.

The spatial distribution of *ACC*, *CSI*, *HSS*, *HK* and *ETS* has an obvious positive correlation with accumulated precipitation. According to Figure 11(c) and Figure 11(d), the QPE algorithm correctly estimated that the precipitation pixels accounted for a relatively high proportion in the three heavy precipitation centres and the northwest mountainous area, indicating that for heavy convective precipitation and long-lasting stratigraphic clouds precipitation, the QPE algorithm's ability to accurately distinguish between precipitation and non-precipitation pixels is stronger. The higher value positions in Figure 11(e), Figure 11(f) and Figure 11(g) are basically the same as Figure 11(c) and (d). However, for areas with heavy precipitation in the northeast, the values of *HSS*, *HK* and *ETS* are relatively low. This is because the heavy precipitation area

in the northeast has higher precipitation frequency during the study period (Figure 10(b)), which makes the random estimation improve the probability of accurately estimating the precipitation pixels, then the advantage of *ACC* and *CSI* over random estimation will be weakened, this results in low *HSS* and *ETS* values. The higher precipitation frequency in the northeast leads to a decrease in the number of imagery pixels identified by both the stations and the QPE algorithm as non-precipitation. this results in high *POFD* values. When the *POD* remains unchanged (Figure 11(a)), the value of *HK* decreases. Comparing Figure 11(e) and Figure 11 (f), the heavy precipitation area in the northeast basically satisfies *HSS = HK*, which means that the algorithm in the northeast is an unbiased estimation . In summary, although the values of *HSS*, *HK*, and *ETS* in the northeast are relatively low, the algorithm still has a high ability to distinguish between precipitation and non-precipitation pixels.

For Figure 11(h), there are 2660 stations with *R* > 0.8, accounting for 59.45%, and 600 stations with *R* < 0.6, accounting for 14.41%. Comparing Figures 11(b) to Figures 11(g), the stations with lower *R* have relatively higher *FAR* and *Bias*, lower *ACC*, *CSI*, *HSS*, *HK* and *ETS*. The 48-h accumulated precipitation of these stations is less than 12.5 mm, and the accumulated precipitation is less than 5 h. This basically means that, during this precipitation process, these stations are in atypical stratus cloud or a convective precipitation process, and the precipitation efficiency is extremely low. For non-precipitation areas in a heavy precipitation process, the QPE algorithm tends to judge them as weak precipitation areas, this shows that the algorithm overestimations the precipitation areas. Although the QPE algorithm tends to overestimate the weak (non-) precipitation area and underestimate the heavy precipitation area, the absolute error for the underestimated heavy precipitation area is much larger than the overestimated weak (non-) precipitation area. Therefore, the spatial distribution of *RMSE* in Figure 11(j) is highly consistent with the spatial distribution of 48-h accumulated precipitation in Figure 2(c). The stations with an *RMSE* greater than 1.6 mm/h in Figure 11(j) are basically connected together, and their coverage is basically the same as the area covered by the stations with a 48-h accumulated precipitation exceeding 50 mm/h in Figure 2(c).

In summary, the estimated ability of the QPE algorithm is as follows: (1) For convective precipitation or stratus convective mixed precipitation with long duration, high intensity and high efficiency, such as in Guangzhou and nearby urban areas in Case 3, the algorithm has high *POD, ACC, CSI, HSS, HK and ETS*, low *FAR*, and *R* and *Bias* are close to 1. Its *RMSE* is affected by the precipitation intensity. In this precipitation process, the QPE algorithm has the strongest estimated ability. (2) For precipitation with long duration and weak intensity per unit time, such as the northwest mountainous area in Case 3, the *POD*, *FAR*, *ACC*, *CSI*, *HSS*, *HK* and *ETS* of the algorithm show roughly the same characteristics as the previous type of precipitation. The *R* also decline slightly, but is still near 0.8. The *RMSE* is greatly reduced. The estimatedability of the QPE algorithm is second to the previous precipitation process. (3) When the duration of precipitation is short and the intensity is only light to moderate rain, such as in the southwest coastal area in Case 3, the *FAR* and *Bias* estimated by the QPE algorithm is relatively high, the *ACC*, *CSI*, *HSS*, *HK*, *ETS* and *R* is relatively low, and the reliability of the estimated ability of the QPE algorithm is low. For Case 1 and Case 2, the hourly spatial distribution of evaluation indicators for both QPE algorithms at each station is shown in Figure S7 of the supporting information.

Figure 12 shows the time series of evaluation indicators of the QPE algorithm for all stations at each time. The red lines represent the average values of the evaluation indicators, from which we can see that the average values of *POD*, *FAR*, *ACC*,

*CSI*, *HSS*, *ETS*, *R*, *Bias* and *RMSE* are 0.97、0.60、0.64、0.40、0.31、0.47、0.19、0.70、2.87 and 1.90mm/h, respectively. From 01:00 to 14:00 on June 24, the *POD*, *ACC*, *CSI*, *HSS*, *HK*, *ETS*, *R* and *RMSE* are high, the *FAR* is low, and the *Bias* tends to 1. According to Figures 6(j) to 6(p) and Figures 8(q) to 8(w), the intensity of the precipitation centre during this period exceeds 16 mm/h, reaching rainstorm level. At this point, the estimation accuracy of the QPE algorithm is strong. Not only is the effect of the evaluation indicator good, but also the intensity of the precipitation centre and the precipitation range of these periods fit with high accuracy, as can be seen by comparing Figure 5 with Figure 6(j–l) and Figure 7 with Figure 8(q–w). This proves that, for the strong convective process with a short precipitation duration and the precipitation intensity reaching rainstorm level, the evaluation indicators show similar characteristics to the first type of precipitation above. At the same time, this means that when the precipitation intensity is large enough, the precipitation duration is no longer the main factor influencing the estimated ability of the QPE algorithm. At 00:00–04:00 on June 23, the *POD*, *ACC*, *CSI*, *HSS*, *HK*, *ETS*, *R* and *RMSE* are higher than their average values, while the *FAR* and *Bias* are lower than their average values. The characteristics of each evaluation indicator during this period are similar to the second type of precipitation above. Comparing Figure 7 with Figure 8(a–d), the precipitation intensity and range are basically successfully fitted, but the estimation of precipitation in localized areas is not good. This verifies the previous conclusion that the QPE algorithm's estimated ability for long-duration and weak-intensity precipitation is inferior to that of strong convective precipitation. From 17:00 to 20:00 on June 23, the *POD*, *ACC*, *CSI*, *HSS*, *HK*, *ETS*, *R* and *RMSE* are all lower than their average values, while *FAR* and *Bias* are higher than their average values. According to Figures 8(i) to 8(l), the precipitation duration during this period is short, the precipitation intensity is weak, and the precipitation process and characteristics are similar to the third type of precipitation above. For Case 1 and Case 2, the time series of evaluation indicators of the QPE algorithm for all stations at each time are shown in Figure S8 of the supporting information.

## 4 Conclusions and discussions

In this study, a machine-learned regression model was established using the RF method to derive QPE from FY-4A observations, in conjunction with cloud parameters and physical quantities. The cross validation results indicate that both DQPE and NQPE RF algorithms performed well in estimating QPE, with the *Bias*, *R* and *RMSE* of DQPE (NQPE) of 2.17 (2.42), 0.79 (0.83) and 1.77mm/h (2.31mm/h), respectively. Overall, the algorithm has a high accuracy in estimating precipitation under heavy rain level or below. Nevertheless, the positive bias still implies an overestimation of precipitation by the QPE algorithm, in addition to certain misjudgements from non-precipitation pixels to precipitation events. Also, the QPE algorithm tends to underestimate the precipitation at the rainstorm or even above levels. Compared to single-sensor algorithm, the developed QPE algorithm can better capture the spatial distribution of land-surface precipitation, especially the centre of strong precipitation. Marginal difference between the data accuracy over sites in urban and rural areas indicate that the model performs well over space and has no evident dependence on landscape.

Further investigations revealed that the estimation accuracy of the QPE algorithm is mainly affected by the rain rate and precipitation duration. More specifically: (1) With respect to strong precipitation at a high rain rate, the QPE algorithm has a

high estimation accuracy, regardless of the duration. Nevertheless, the *RMSE* is mainly affected by the rain rate, implying when the rain rate is large enough (rainstorm level or above), the precipitation duration is no longer a factor affecting the accuracy of the QPE algorithm. (2) For precipitation processes with a long duration at a low rain rate, the algorithm shows a roughly similar accuracy as the previous type of precipitation, but with a slightly degradation in *R* whereas a significant increase in *RMSE*. (3) For precipitation with short duration at a small to moderate intensity, the QPE algorithm showed a relatively high *FAR* and *Bias* albeit a relatively low *ACC*, *CSI*, *HSS*, *HK*, *ETS* and *R,* implying a low accuracy of the QPE algorithm.

In general, by synergistically using high-density automatic station data and meteorological physical quantity fields, the QPE algorithm developed in this study provides a promising way for quantitative estimation of summer precipitation over East Asia from FY-4A satellite observations. Our results also highlight the beneficial effect of satellite cloud parameters and meteorological physical variables that were neglected in previous studies in improving the estimation accuracy of QPE. Moreover, by replacing the ERA5 reanalysis data that were used in this study with meteorological forecast fields such as global forecast system from China T639, ECMWF or GFS, this RF model framework can be easily adapted to quantitatively estimate QPE in near real-time.

## Code availability

The model in this paper is based on the random Forest data package in the R language, and our implementation and analysis code are available upon request to the corresponding author (yyj1985@nuist.edu.cn).

## Data availability

All Fengyun-4 satellite data used in this paper can be downloaded from China National Meteorological Satellite Centre at http://www.nsmc.org.cn/NSMC/Home/Index.html. ERA5 reanalysis data is from the Copernicus Climate Change Service at https://cds.climate.copernicus.eu/cdsapp#!/dataset/reanalysis-era5-single-levels?tab=form. DEM data is from Geospatial Data Cloud managed by Computer Network Information Centre Chinese Academy of Sciences at http://www.gscloud.cn. The data of high-density automatic stations are not available to the public. Please direct any inquiries regarding the data to the corresponding author (yyj1985@nuist.edu.cn).

## Supplement

The supplement related to this paper is available online at: Supplement.

## Author contributions

Conceptualization, Y.Y.; methodology, X.L. and Y.Y.; software, X.L.; validation, G.N. and Y.Y.; formal analysis, X.L., and Y.Y.; data curation, X.L., Z.L., and W.L.; writing—original draft preparation, X.L.; writing—review and editing, X.B., Y.Z., Z.H., C.L., and Y.Y.; visualization, X.L. and J.M.; supervision, Y.Y.; funding acquisition, Y.Y. All authors have read and agreed to the published version of the manuscript.

## Competing interests

The authors declare that they have no conflict of interest.

### Acknowledgements

The author would like to thank Yali Chao of Sun Yat-sen University helped in drawing the distribution map of high-density automatic stations.

### Financial support

This research has been supported by the National Key Research and Development Program of China (grant no.2018YFC1506502), the National Natural Science Foundation of China (grant no. 42175098), College Students' Innovative Entrepreneurial Training Plan Program of Jiangsu China (grant no. 202010300057Y).

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

**Table 1: Variables used in the QPE algorithm.**

| | Variables |
|---|---|
| Geographic information | Longitude, Latitude, DEM |
| Channel information of AGRI | T0.47*, T0.65*, T0.825*, T1.375*, T1.61*, T2.25*, T3.75H, T3.75L, T6.25, T7.1, T8.5, T10.7, T12.0, T13.5 |
| Combined information of AGRI | T6.25–T10.7, T8.5–T10.7, T7.1–T12.0, T12.0–T10.7, T3.75L–T7.1, T3.75L–T10.7, $CTT_G$ |
| Cloud parameters of AGRI | CTT, CTH, CLT, CLP |
| ERA5 | ISOR, CAPE, EWSS, K-Index, TCW, TCWV |

Notation: asterisk (*) indicates that the variable does not appear in the NQPE algorithm.

**Table 2: Evaluation metrics used in this study.**

| Evaluation metric | Equation |
|---|---|
| Bias | $Bias = \dfrac{A + B}{A + C}$ |
| POD | $POD = \dfrac{A}{A + C}$ |
| FAR | $FAR = \dfrac{B}{A + B}$ |
| ACC | $ACC = \dfrac{A + D}{A + B + C + D}$ |
| CSI | $CSI = \dfrac{A}{A + B + C}$ |
| HSS | $HSS = \dfrac{ACC - A_{ref1}}{1 - A_{ref1}}, \quad A_{ref1} = \dfrac{(A + B)(A + C) + (C + D)(B + D)}{A + B + C + D}$ |
| HK | $HK = POD - POFD = \dfrac{AD - BC}{(A + C)(B + D)}, \quad POFD = \dfrac{B}{B + D}$ |
| ETS | $ETS = \dfrac{CSI - \dfrac{A_{ref2}}{A + B + C}}{1 - \dfrac{A_{ref2}}{A + B + C}} = \dfrac{A - A_{ref2}}{A + B + C - A_{ref2}}, \quad A_{ref2} = \dfrac{(A + B)(A + C)}{A + B + C + D}$ |
| R | $R = \dfrac{\sum_{i=1}^{n}(G_i - \bar{G})(S_i - \bar{S})}{\sum_{i=1}^{n}(G_i - \bar{G}) \sum_{i=1}^{n}(S_i - \bar{S})}$ |
| RMSE | $RMSE = \sqrt{\dfrac{1}{n}\sum_{i=1}^{n}(G_i - S_i)^2}$ |

Notation: $A$ is the number of imagery pixels identified by both the stations and the QPE algorithm as precipitation; $B$ is the number of imagery pixels identified by the QPE algorithm as precipitation but not by the stations; $C$ is the number of imagery pixels identified by the stations as precipitation but not by the QPE algorithm; $D$ is the number of imagery pixels identified by both the stations and the QPE algorithm as non-precipitation. $G_i$ is the precipitation observed by stations; $S_i$ represents the precipitation estimated by the QPE algorithm.

**Table 3: Evaluation metrics in training set and testing set of DQPE (NQPE) algorithm**

| Model name | POD | FAR | ACC | CSI | HSS | HK | ETS |
|---|---|---|---|---|---|---|---|
| Training set of DQPE | 1.00 | 0.43 | 0.78 | 0.57 | 0.57 | 0.69 | 0.39 |
| Testing set of DQPE | 0.98 | 0.55 | 0.65 | 0.45 | 0.37 | 0.50 | 0.23 |
| Training set of NQPE | 1.00 | 0.49 | 0.76 | 0.51 | 0.51 | 0.67 | 0.35 |
| Testing set of NQPE | 0.98 | 0.59 | 0.63 | 0.40 | 0.33 | 0.49 | 0.20 |

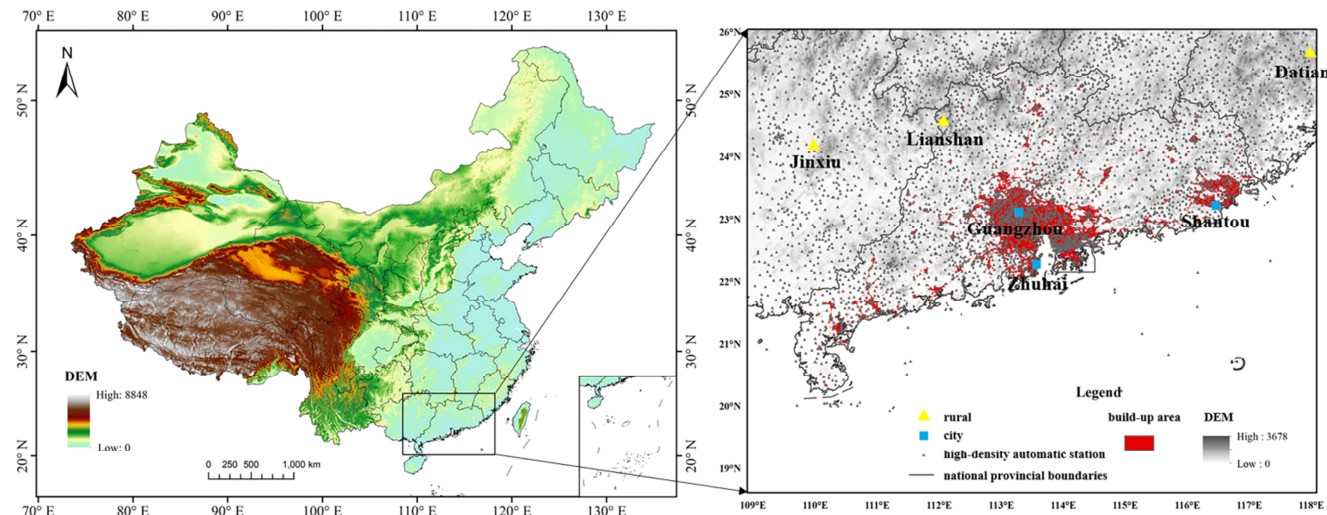

**Figure 1: Distribution of automatically operated meteorological stations over the study area.**

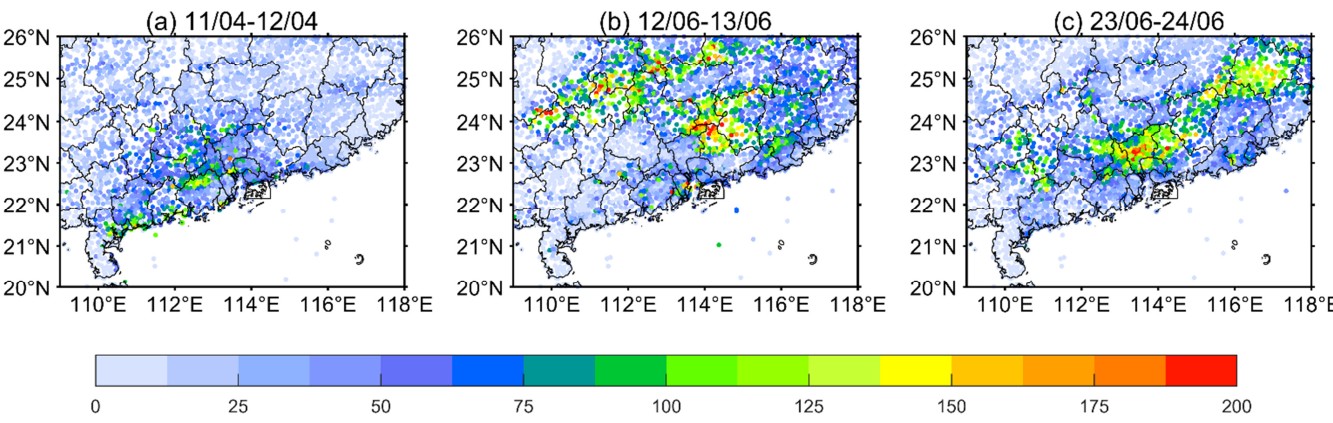

**Figure 2: Spatial distribution of precipitation during the three South China rainstorms: (a) April 11-12, 2019; (b) June 12-13, 2019; (c) June 23-24, 2019.**

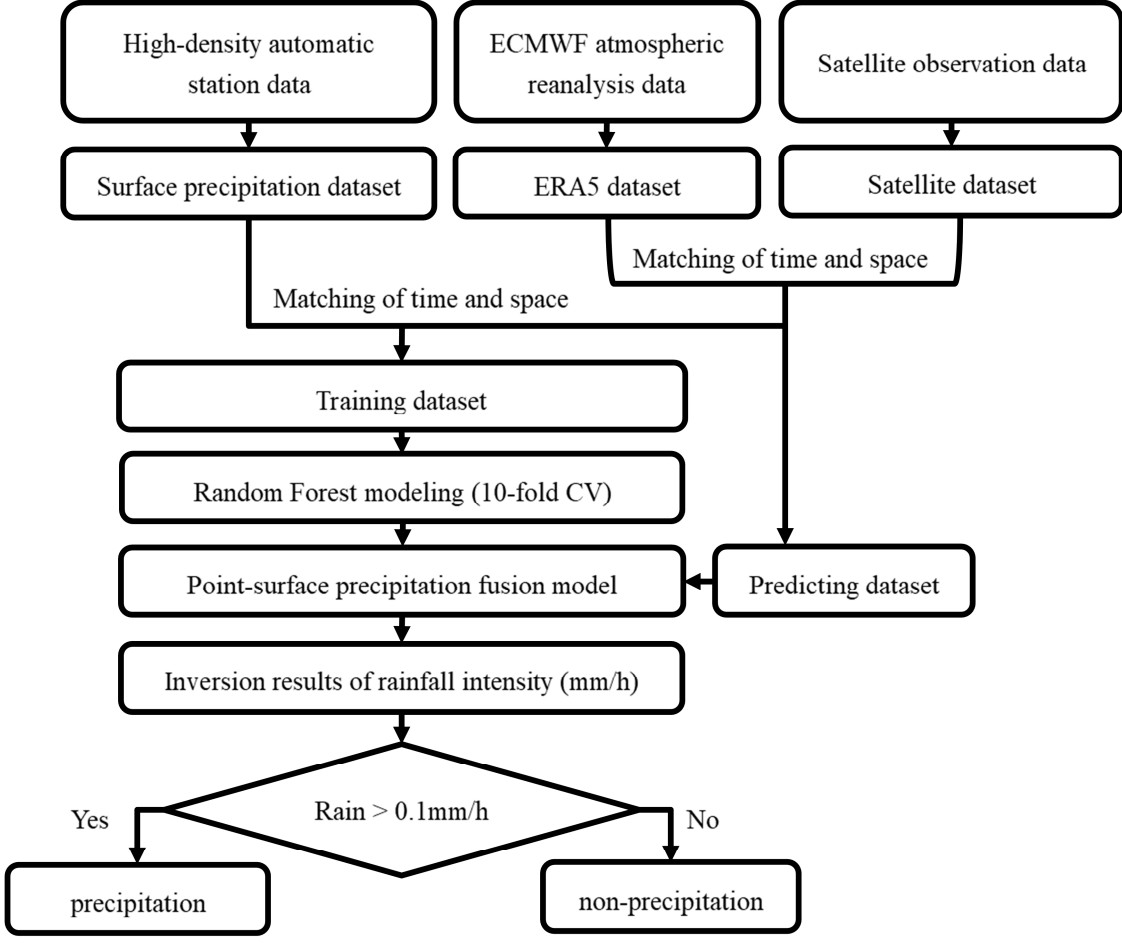

**Figure 3: Flowchart for the QPE algorithm using the RF model.**

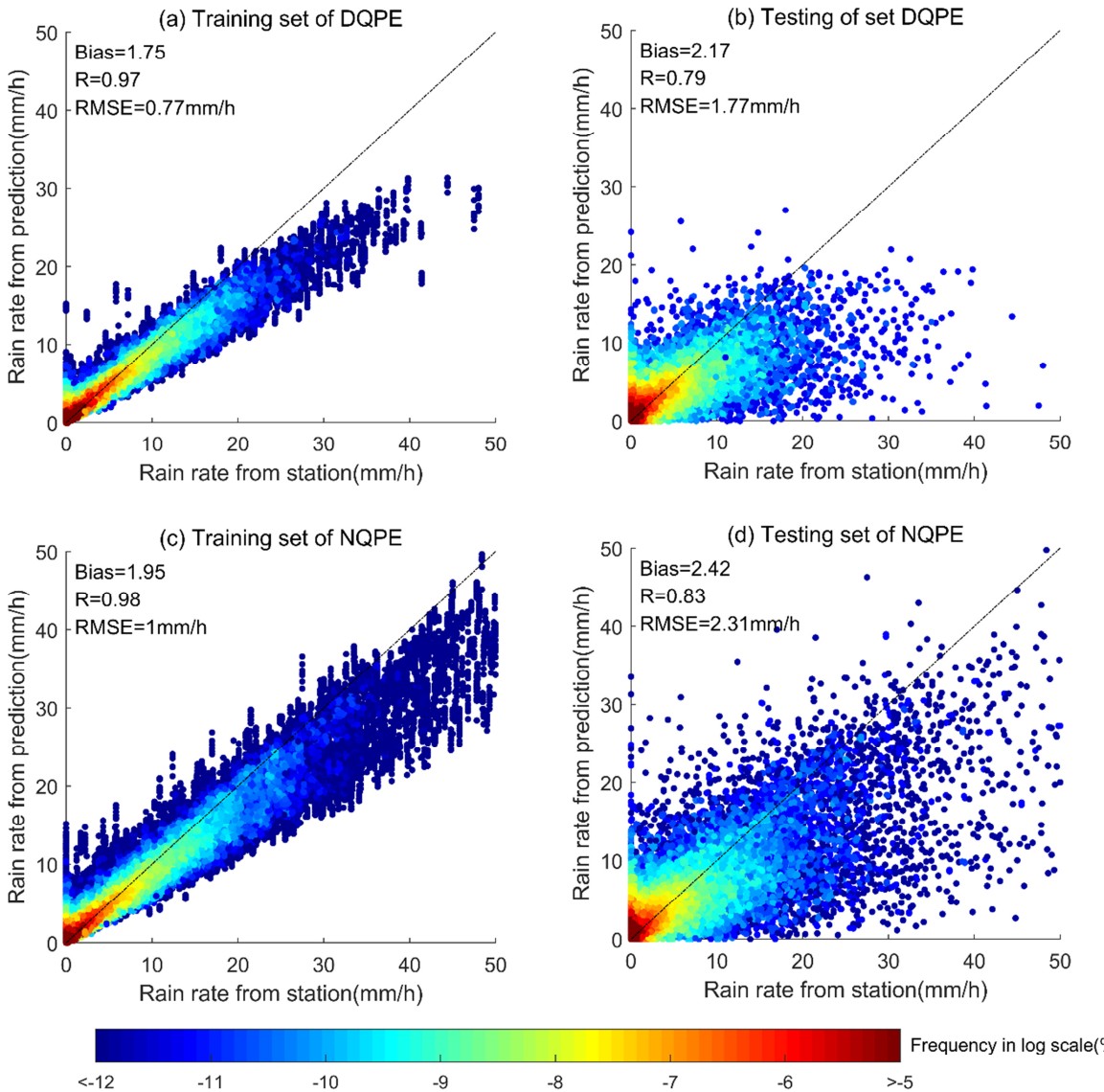

**Figure 4: Comparison of the precipitation measured by high-density automatic stations and that** estimated **by the QPE algorithm: (a) training set of DQPE; (b) testing set of DQPE; (c) training set of NQPE; (d) testing set of NQPE. Color bar: occurrence frequency (on a log scale) at intervals of 0.5 mm/h.**

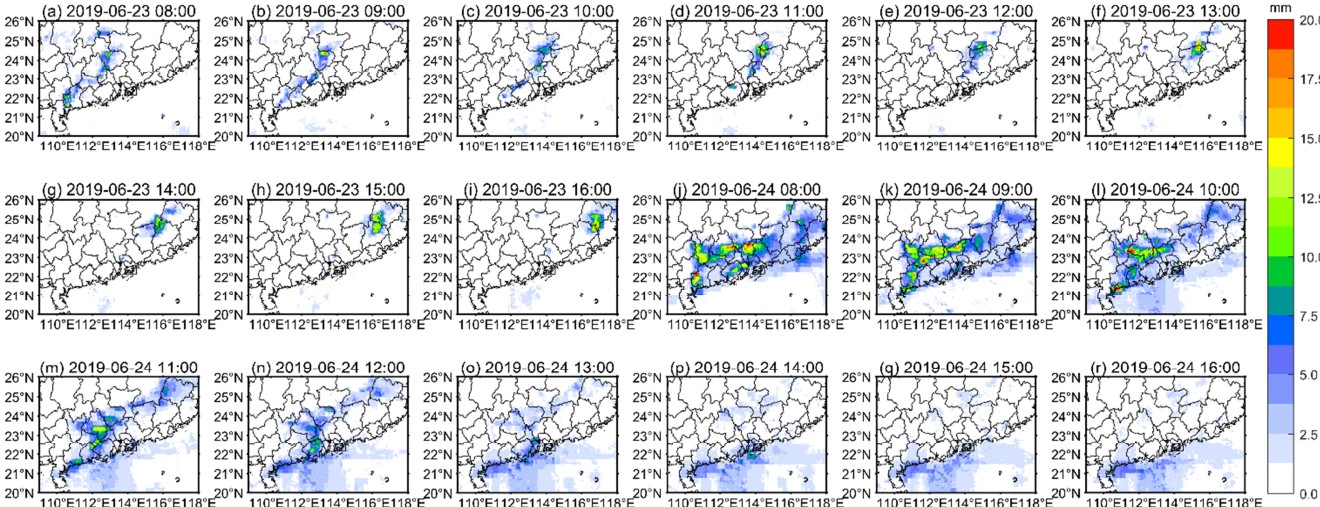

**Figure 5: Estimated precipitation of the DQPE algorithm at (a–i) 0800–1600 BJT on June 23; (j–r) 0800–1600 BJT on June 24.**

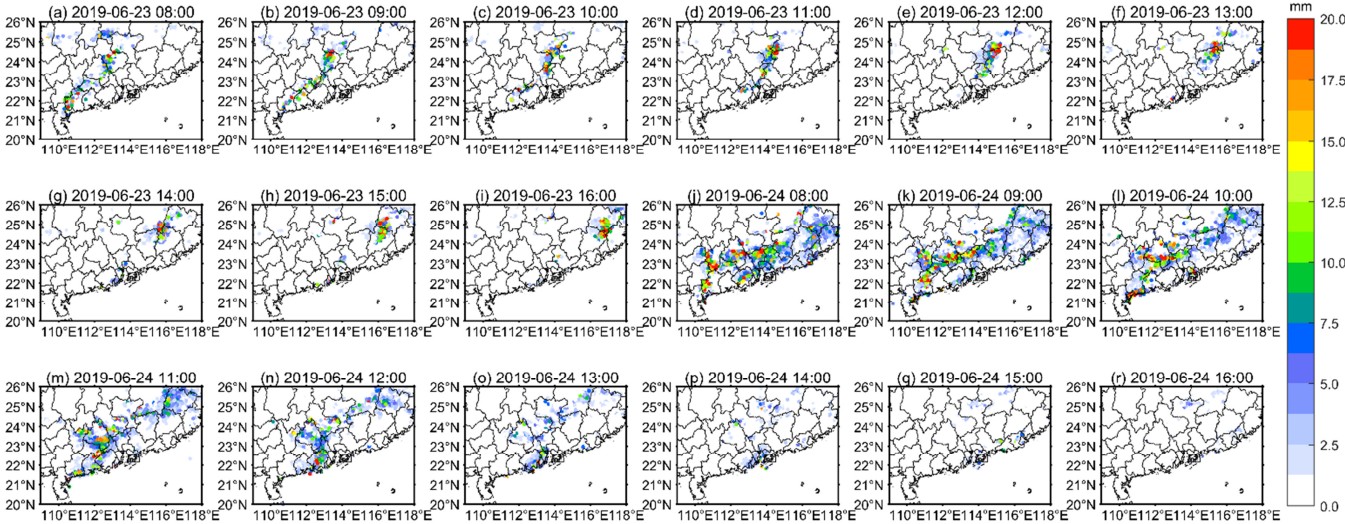

**Figure 6: Actual precipitation based on high-density automatic stations at (a–i) 0800–1600 BJT on June 23; (j–r) 0800–1600 BJT on June 24.**

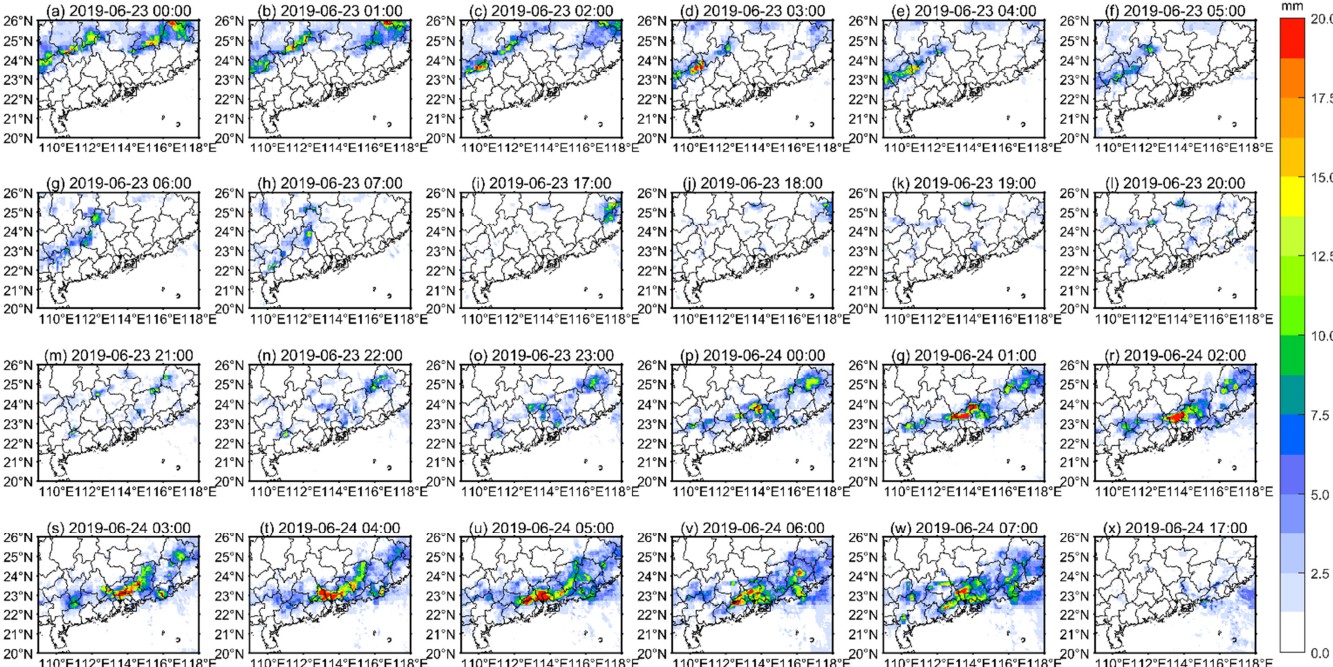

**Figure 7: Estimated precipitation of the NQPE algorithm at (a–h) 0000–0700 BJT on June 23, (i–o) 1700–2300 BJT on June 23, (p–w) 0000–0700 BJT on June 24, and (x) 1700 BJT on June 24.**

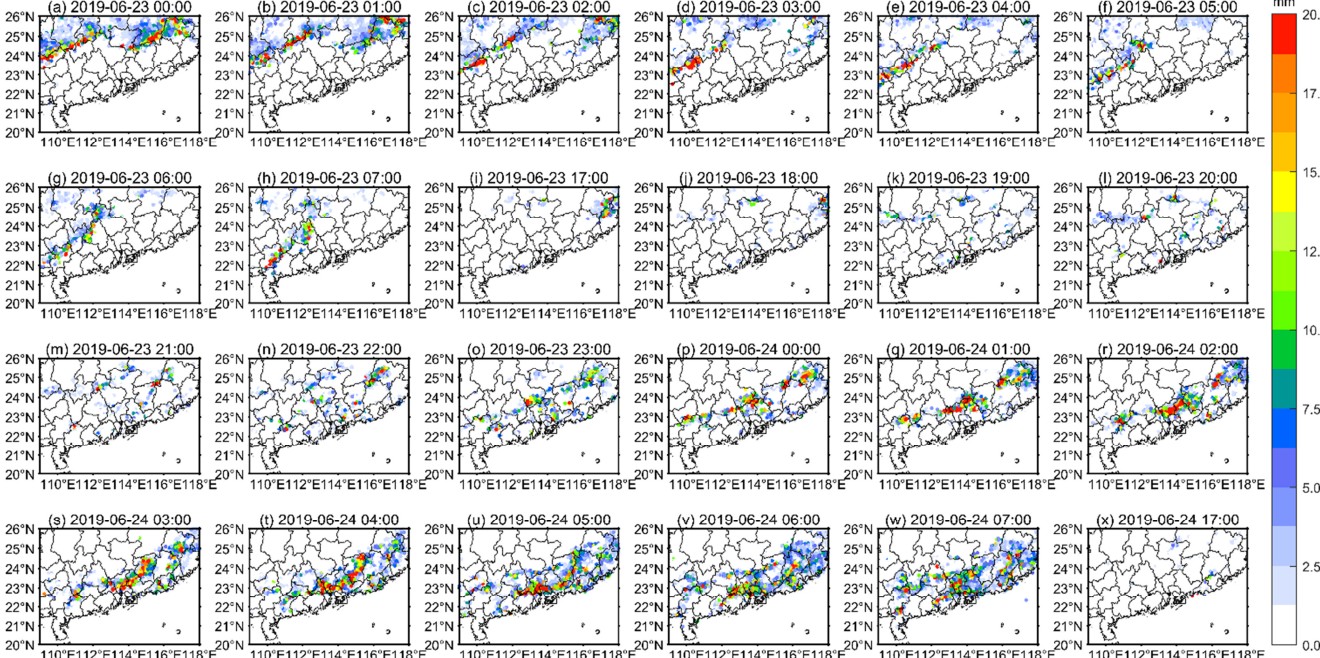

**Figure 8: Actual precipitation based on high-density automatic stations at (a–h) 0000–0700 BJT on June 23, (i–o) 1700–2300 BJT on June 23, (p–w) 0000-0700 BJT on June 24, and (x) 1700 BJT on June 24.**

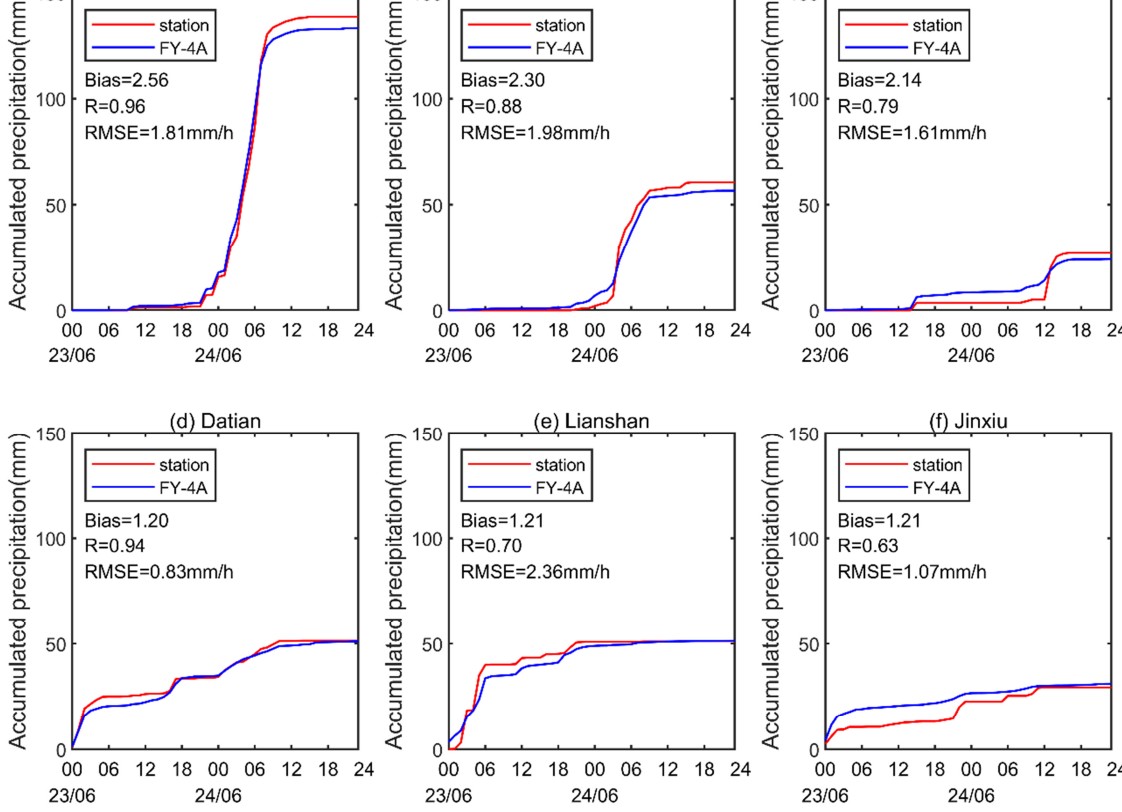

**Figure 9: Accumulated precipitation in different areas: (a–c) city stations; (d–f) rural stations.**

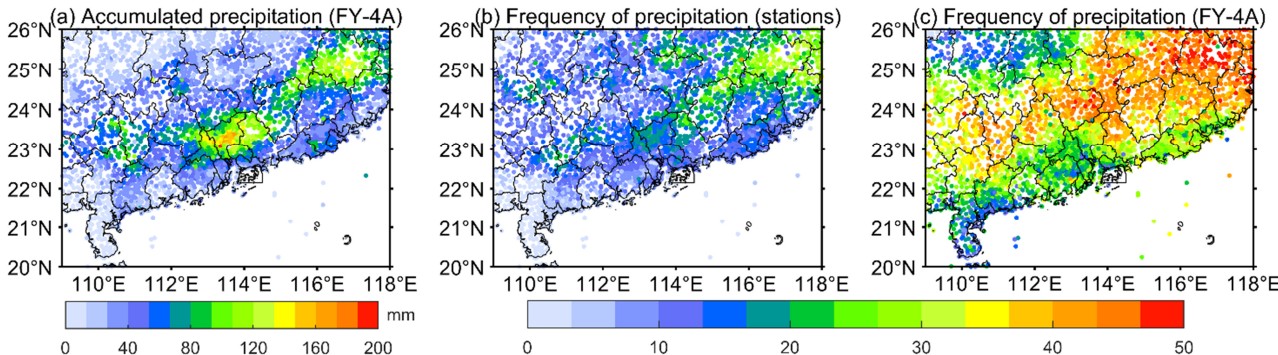

**Figure 10: Spatial distribution of accumulated precipitation: (a) accumulated precipitation estimated by the QPE algorithm; (b) actual precipitation frequency observed by high-density automatic stations; (c) precipitation frequency estimated by the QPE algorithm.**

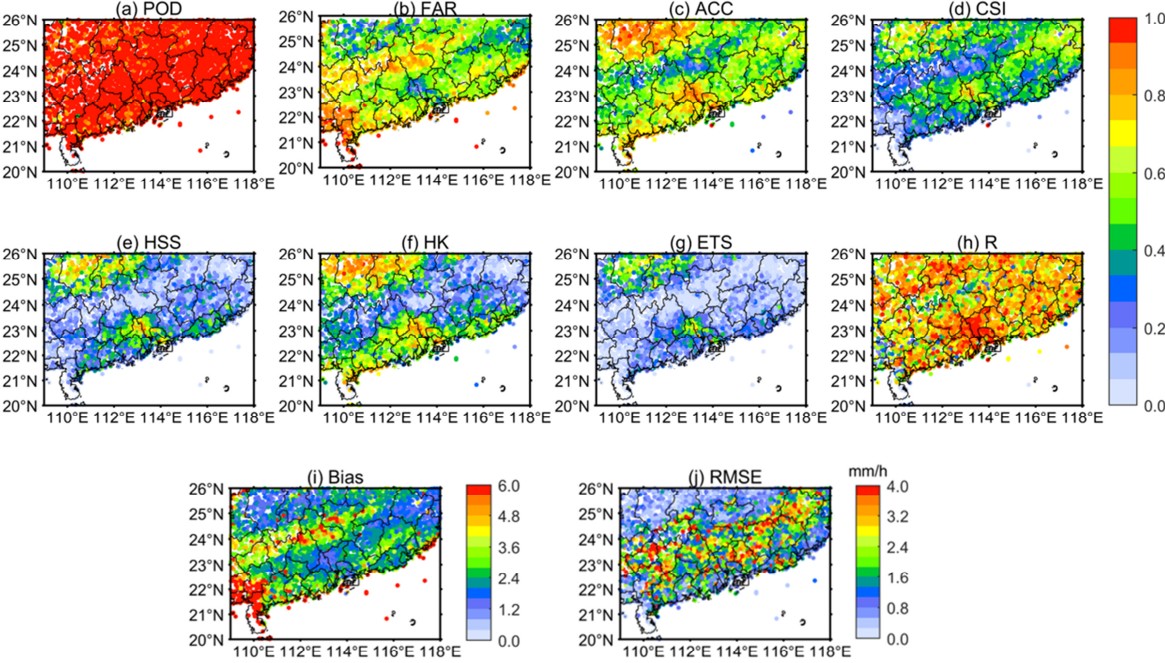

**Figure 11: Spatial distribution of evaluation indicators of the QPE algorithm for all stations: (a) *POD*; (b) *FAR*; (c) *ACC*; (d) *CSI*; (e) *HSS*; (f) *HK*; (g) *ETS*; (h) *R*; (i) *Bias*; (j) *RMSE*.**

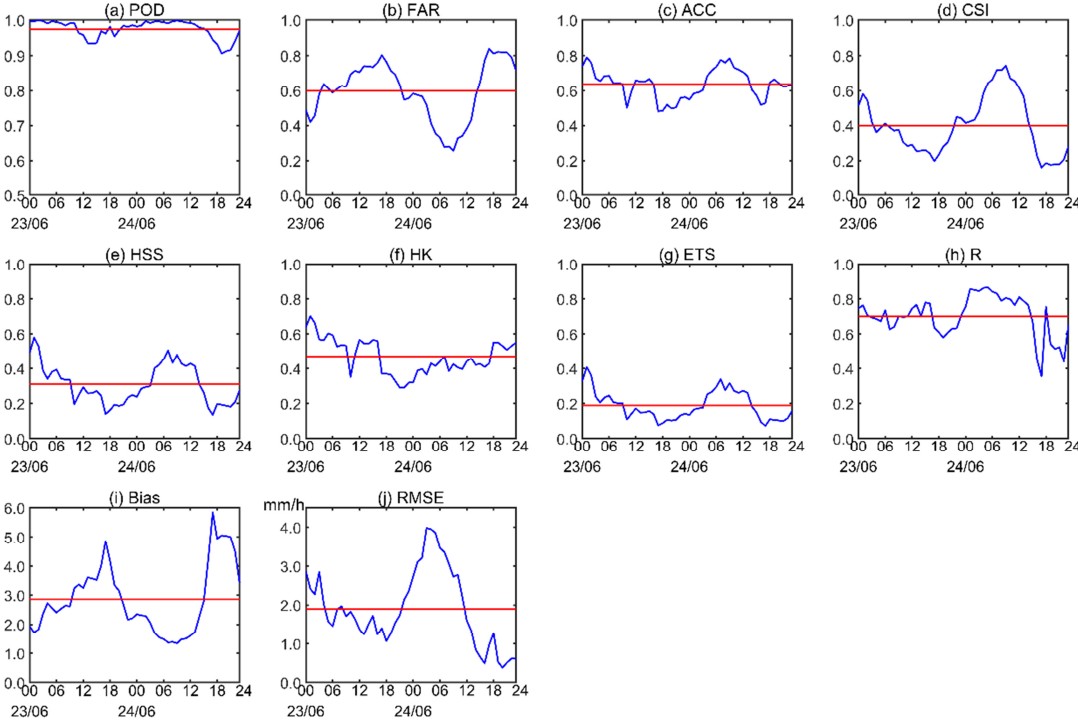

**Figure 12: Time series of evaluation indicators of the QPE algorithm for all stations at each time: (a) *POD*; (b) *FAR*; (c) *ACC*; (d) *CSI*; (e) *HSS*; (f) *HK*; (g) *ETS*; (h) *R*; (i) *Bias*; (j) *RMSE*.**

