# Peer review of "Leveraging machine learning for quantitative precipitation estimation from Fengyun-4 geostationary observations and ground meteorological measurements"

_Atmospheric Measurement Techniques, 2021_

## Author Comment (AC1)

**Reponses to referee(s) comments**

**Reviewer 1**

The paper "Leveraging machine learning for quantitative precipitation estimation from Fengyun-4 geostationary observations and ground meteorological measurements", by Li and co-workers, presents a preliminary application of a machine learning technique to retrieve precipitation hourly rate from geostationary VIS-IR data. A Random Forest classificator is applied to multispectral AGRI data on board the Chinese FY-4 satellite for 3 2-day storms occurred in Southern China: calibration and validation of the estimates are performed against hourly automatic weather station data.

The paper is interesting since there is very little published work on FY-4 data, however, I think the present manuscript needs a deep revision before to be published on AMT. Below, my suggestion to improve the quality of the manuscript.

**RESPONSE: Thank you for giving us the opportunity to improve the quality of this manuscript. We have substantially revised this manuscript by following your insightful comments and constructive suggestions. Please find out our point-by-point responses below.**

Introduction

Lines 51 and following: any introduction on multisensor precipitation estimation cannot forget international programmes that provides high quality and high resolution precipitation products at global or continental scale, such as NASA-GPM or H-SAF. Please, complete.

**RESPONSE: Thanks for your valuable suggestion. More relevant literature has been reviewed and cited in the introduction section to bridge the readership gap, please see the added contents at lines 52-57 in our revised manuscript:**

**"……,and the Passive Microwave Neural Network Precipitation Retrieval approach for The EUMETSAT Satellite Application Facility on Support to Operational Hydrology and Water Management (H-SAF) (Mugnai et al., 2013; Sanò et al., 2014, 2018), as well as active precipitation retrieval methods based on the Precipitation Radar (PR) carried onboard the Tropical Precipitation Measuring Mission satellite (Iguchi et al., 2000) and The Global Precipitation Measurement (GPM) Core Observatory spacecraft (Sharifi et al., 2016; Tan and Duan, 2017). "**

Line 85, and in many other parts of the paper, are mentioned high-density stations, without any quantitative indication on how the density is measured and how "high density" is defined. Please, give more quantitative details on the station distribution.

**RESPONSE: Thanks for pointing out. More detailed descriptions have been provided to better introduce automatically operated meteorological stations. Please refer to lines 96-98 in the revised manuscript:**

**"In addition to 215 national operated meteorological stations, there are also 4,706 automatic stations over the study region, with a mean distance between them less than 10 km (Figure 1). Also, stations are deployed with higher density in the urban built-up area with relative to**

the rural area."

Line 96. In Figure 1 please write the meaning of red areas (NPP_NTL?).
**RESPONSE: Thanks for your constructive comment. NPP_NTL represents nighttime light (NTL) data obtained by the Visible Infrared Imaging Radiometer Suite (VIIRS) on board the Suomi National Polar Orbiting Partnership (NPP). We used this dataset to extract urban built-up areas in South China. To ease the readership, we have changed NPP_NTL to build-up area in the legend of Figure 1. We also explained this at lines 101-103 in the revised manuscript:**
**"The red shading area in Figure 1 shows the build-up area in South China, which was extracted using nighttime light (NTL) data obtained by the Visible Infrared Imaging Radiometer Suite (VIIRS) on board the Suomi National Polar Orbiting Partnership (NPP) satellite."**

[Figure]

**Figure 1: Distribution of automatically operated meteorological stations over the study area.**

Lines 106 and line 108 mention levels: "met the levels for large-scale heavy precipitation" and "met the heavy rain level". How are these levels defined?
**RESPONSE: Thanks for your constructive comments. The rainfall levels used in this paper is defined according to the "Grade of Precipitation", a Chinese national standard implemented on August 1, 2012. In this standard, rainfall is divided into seven grades: light rainfall, light rain, moderate rain, heavy rain, rainstorm, heavy rainstorm and extraordinary rainstorm. Detailed thresholds for rainfall classification were supplied in table S1 in the supplementary information. Since it is challenging to determine the coverage area of quantitative precipitation simply based on site-based precipitation data, so we deleted the qualitative description of "large-scale" in the revised manuscript.**

**Table S1 Classification of rainfall levels in different periods (Unit: mm)**

| Level | Rainfall in different periods | |
|---|---|---|
| | Rainfall in 1 hours | Rainfall in 24 hours |
| light rainfall | < 0.1 | < 0.1 |
| Light rain | 0.1 ~ 1.5 | 0.1 ~ 9.9 |
| Moderate rain | 1.6 ~ 6.9 | 10.0 ~ 24.9 |
| Heavy rain | 7.0 ~ 14.9 | 25.0 ~ 49.9 |
| Rainstorm | 15.0 ~ 39.9 | 50.0 ~ 99.9 |
| Heavy rainstorm | 40.0 ~ 49.9 | 100.0 ~ 249.9 |
| Extraordinary rainstorm | ≥50.0 | ≥250.0 |

Data

Line 128, very likely, 4km is the nominal resolution at nadir.

**RESPONSE: Thanks for your kind suggestion. Indeed, 4km is the nominal resolution of FY-4A/AGRI at nadir. We have corrected this at lines 140-141 in the revised manuscript: "FY4A/AGRI provides level-1 dataset with resolutions of 500 m, 1 km, 2 km and 4 km at nadir, and 4 km at nadir for level-2 dataset."**

Lines 145-154. ERA5 fields come with significant latency (5 days for the "preliminary daily updates" and 3 months for the "quality-assured data". Are these times compliant to the Authors' aim to "monitor flood" (line 81)? Moreover, how could be possible the "the real-time monitoring and prediction of summer precipitation over East Asia" (lines 383-384)? Please discuss the temporal applicability of the proposed technique.

**RESPONSE: Thanks for your insightful comments. The main goal of this paper is to develop a random forest model framework to estimate QPE from FY-4A. We agree with you that there is large time lag for ERA5 fields. However, for practical applications of the RF model framework in the future, instead of ERA5 reanalysis data, the meteorological physical quantity forecast fields of the global forecast system from China T639, ECMWF or GFS can be combined with real-time FY-4A satellite spectral information and high-density automatic station observations to quantitatively estimate real-time, large-scale, and dynamic continuous precipitation over East Asia. Therefore, our proposed FY-4A QPE algorithm has important potentials and broad application value for precipitation monitoring in real-time, as well as rainstorm disaster prevention and reduction. We have added the above discussion in the last section.**

**Sorry for the misunderstanding on the lines 81, and 383-384. We have revised it accordingly, and reorganized as followings:**

**"……, the FY-4A QPE algorithm established in the present work offers important scientific support and application value for the real-time monitoring and prediction of summer precipitation over East Asia" has been changed to "……, the FY-4A QPE algorithm established in the present work offers important scientific support and application value for quantitative estimation of summer precipitation over East Asia"**

Lines 155-171. Please, improve this description. First try to clearly separate different steps of the algorithm (e.g. with bullet points), then use different fonts to define variables in the text (mtray, ntray...).

**RESPONSE: We are grateful to your valuable comments. We have rewritten this section to ease the readership. We have corrected this at lines 171-227 in the revised manuscript:**

**"A data-driven regression model was established between the observed precipitation and satellite data as well as cloud parameters using the RF method. The essence of the RF data estimation model is as follows:**

1) **The input variables to the RF model are shown in Table 1, including geographic information, channel information, combined channel information, cloud parameter products, and ERA5 data. A Daytime Quantitative Precipitation Estimate (DQPE) algorithm and a Nighttime Quantitative Precipitation Estimate (NQPE) algorithm were constructed separately,due to different input variables between daytime and nighttime. The DQPE algorithm is used to estimate the precipitation from 8:00 to 16:00, and the NQPE algorithm is used to estimate the remaining time periods. The visible light channel at nighttime cannot produce valid observational information, so the NQPE algorithm removes these variables. The CTT gradient($CTT_G$) in the combined channel information is closely related to the rain rate, defined as follows Eq. (1):**

[revised manuscript text omitted]

Line 184 and elsewhere. Please, do not use the word "prediction" here and in the whole document to refer to the output of your algorithm, use "estimate", instead.

**RESPONSE: Thanks for your suggestion. It has been corrected in the revised manuscript.**

Lines 195-200. Here is the main lack of the paper: POD and FAR cannot be used separately to assess the quality of an estimates. Besides an error in the sentence ("optimal value of FAR is 1, and the worst value is 0", actually, the opposite is true), to measure the capability of the technique to correctly classify wet/dry pixel you need or to comment both POD and FAR number together (and avoid sentences as on line 17), or to compute synthetic indicators such as Equitable Threat Score (ETS), Hanssen and Kuiper or Heidke Skill Score, and do again the analysis looking at the values of these indicators as reference.

**RESPONSE: Thank you so much for your constructive comments. Per your suggestions, we have defined ten indicators to evaluate the accuracy of the QPE algorithm in the revised manuscript. In order to quantitatively assess the classification results of precipitation and non-precipitation pixels, we introduced eight classical metrics: bias score (*Bias*, *Bias* = 1 unbiased, *Bias* < 1 underestimation, *Bias* > 1 overestimation), probability of detection (POD, optimal = 1), false alarm ratio (*FAR*, optimal = 0), accuracy (*ACC*, optimal = 1), Critical Success Index (*CSI*, optimal = 1), Heidke Skill Score (*HSS*, optimal = 1), Hanssen and Kuiper (*HK*, optimal = 1), and Equitable Threat Score (*ETS*, optimal = 1). The other two indicators can be used to quantitatively evaluate the accuracy of precipitation estimation based on the QPE algorithm. They are correlation coefficient (*R*, optimal = 1), and root-mean-square error (*RMSE*, optimal = 0). We further evaluated the RF model of QPE by using these indicators in the revision.**

Results

Line 205, figure 4, please, use a reasonable number of digits in the numbers reported on the panels. Moreover, POD and FAR should be <1.

**RESPONSE: Thanks for your constructive suggestion. Due to increasing evaluation indicators, we only show *Bias*, *R* and *RMSE* in Figure 4, and put the remaining evaluation indicators in the Table 3.**

**Table 3: Evaluation metrics in training set and testing set of DQPE (NQPE) algorithm**

| Model name | POD | FAR | ACC | CSI | HSS | HK | ETS |
|---|---|---|---|---|---|---|---|
| Training set of DQPE | 1.00 | 0.43 | 0.78 | 0.57 | 0.57 | 0.69 | 0.39 |
| Testing set of DQPE | 0.98 | 0.55 | 0.65 | 0.45 | 0.37 | 0.50 | 0.23 |
| Training set of NQPE | 1.00 | 0.49 | 0.76 | 0.51 | 0.51 | 0.67 | 0.35 |
| Testing set of NQPE | 0.98 | 0.59 | 0.63 | 0.40 | 0.33 | 0.49 | 0.20 |

Lines 214-215. This sentence is a speculation not supported by evidence, please motivate it better or cancel.

**RESPONSE: Thanks for your suggestion. We have deleted this sentence.**

Lines 218-221. To better illustrate this issue, please, use the indicators I suggested few lines above (ETS, HK…)

**RESPONSE: Thanks for your constructive suggestions. We have added new evaluation indicators in the revised manuscript.**

Line 223. Again, an unsupported sentence, please, give evidence or remove.

**RESPONSE: Thanks, it has been removed.**

Lines 224-230. This paragraph not clear: if the Authors have the feeling that the dataset is not large enough to carry on proper training/testing procedure (that was also my feeling at the beginning) why not to add some more case?

**RESPONSE: We are sorry for the misleading. In fact, the number of training set samples during the daytime (nighttime) is 724680 (1230894), and the number of testing set samples is 80520 (136766), which is adequate to initiate a machine learning practice. Our purpose is to explain the notable difference in the performance between testing and training, more specifically, the over-fitting issue. This issue is caused largely due to the complex relationships between the dependent variable and regressors, while the model fails to generalize well on unseen data (e.g., the testing dataset). To avoid misleading, we have rephrased the sentences to ease the readership. Overall, the main objective of this study is to present a machine learning framework that can be applied to estimate QPE from FY-4A with the involvement of meteorological physical quantities. For demonstration, we only employed three typical rainstorm cases in this study to illustrate the capability of the proposed method. A better regression could be established in the future by using data from more diverse cases to improve the generalization capacity of QPE estimation model. Again, thanks for pointing out this flaw.**

Lines 245-246. I do not see this sentence comes out. Numerical indicators (POD, FAR, R, ETS….) tell us much more from the quantitative point of view with respect to simple visual comparisons of rain maps. Please, use numbers if you want to make quantitative assessments.

**RESPONSE: Thanks for your constructive suggestions. We have revised this sentence and added new evaluation indicators in the revised manuscript.**

Lines 263-264. Absolute and relative errors are not defined in the text.

**RESPONSE: Thanks for your suggestion. We rephrased the verification of the QPE results and deleted the part of absolute and relative error.**

Lines 291-293. This sentence does not tell anything about the technique accuracy, since POD alone is considered.

**RESPONSE: Thanks for your kind suggestion. We have cancelled this sentence in the revision.**

In general, discussion and conclusion have to be rewritten once the new indictors I suggested will be implemented.

**RESPONSE: Many thanks for your kind suggestion. We have rewritten the conclusion and discussion section as follows:**

"In this study, a machine-learned regression model was established using the RF method to derive QPE from FY-4A observations, in conjunction with cloud parameters and physical quantities. The cross validation results indicate that both DQPE and NQPE RF algorithms performed well in estimating QPE, with the *Bias*, *R* and *RMSE* of DQPE (NQPE) of 2.17 (2.42), 0.79 (0.83) and 1.77mm/h (2.31mm/h), respectively. Overall, the algorithm has a high accuracy in predicting precipitation under heavy rain level or below. Nevertheless, the positive bias still implies an overestimation of precipitation by the QPE algorithm, in addition to certain misjudgements from non-precipitation pixels to precipitation events. Also, the QPE algorithm tends to underestimate the precipitation at the rainstorm or even above levels. Compared to single-sensor algorithm, the developed QPE algorithm can better capture the spatial distribution of land-surface precipitation, especially the centre of strong precipitation. Marginal difference between the data accuracy over sites in urban and rural areas indicate that the model performs well over space and has no evident dependence on landscape.

Further investigations revealed that the estimation accuracy of the QPE algorithm is mainly affected by the rain rate and precipitation duration. More specifically: (1) With respect to strong precipitation at a high rain rate, the QPE algorithm has a high prediction accuracy, regardless of the duration. Nevertheless, the *RMSE* is mainly affected by the rain rate, implying when the rain rate is large enough (rainstorm level or above), the precipitation duration is no longer a factor affecting the accuracy of the QPE algorithm. (2) For precipitation processes with a long duration at a low rain rate, the algorithm shows a roughly similar accuracy as the previous type of precipitation, but with a slightly degradation in *R* whereas a significant increase in *RMSE*. (3) For precipitation with short duration at a small to moderate intensity, the QPE algorithm showed a relatively high *FAR* and *Bias* albeit a relatively low *ACC*, *CSI*, *HSS*, *HK*, *ETS* and *R*, implying a low accuracy of the QPE algorithm.

In general, by synergistically using high-density automatic station data and meteorological physical quantity fields, the QPE algorithm developed in this study provides a promising way for quantitative estimation of summer precipitation over East Asia from FY-4A satellite observations. Our results also highlight the beneficial effect of satellite cloud parameters and meteorological physical variables that were neglected in previous studies in improving the prediction accuracy of QPE. Moreover, by replacing the ERA5 reanalysis data that were used in this study with meteorological forecast fields such as global forecast system from China T639, ECMWF or GFS, this RF model framework can be easily adapted to quantitatively estimate QPE in near real-time."

---

## Author Comment (AC2)

**Reviewer 2**

The title could be revised as: "Leveraging machine learning for quantitative precipitation estimatio n from Fengyun-4 geostationary observations and ground meteorological measurements" **RESPONSE: Thanks, it has been revised per your suggestion.**

"Large-scale and high-quality precipitation products derived from satellite remote sensing spectral data have always been a challenging issue in satellite quantitative precipitation estimation (QPE). Moreover, QPE research related to China's Fengyun-4A (FY-4A) geostationary satellite is still ve ry limited." could be revised as: "Deriving large-scale and high-quality precipitation products fro m satellite remote sensing spectral data is always challenging in quantitative precipitation estimati on (QPE), and limited studies have been conducted even using the China's latest Fengyun-4A (FY -4A) geostationary satellite."

**RESPONSE:** Thanks, it has been revised.

Line 156: "We constructed an RF model through the RF data package in R language, and establish ed a relationship model of the satellite spectrum, cloud parameters and precipitation for the inversi on and prediction of precipitation" could be changed to "A data-driven regression model was estab lished between the observed precipitation and satellite spectrum as well as cloud parameters using the RF method."

**RESPONSE:** Thanks, it has been revised.

Figure 1: legend notations should be corrected, e.g., all\_station should be automatic station? **RESPONSE: Thanks for pointing out, we have revised them in this revision.**

Figure 4: the results indicate significant over-fitting issue of these two prediction models, what are possible reasons? Also, the high precipitation was underestimated, is there any possible way to ad dress this?

**RESPONSE:** Thanks for your valuable comments. We think that the significant over-fitting issue could be attributable to the high complexity of the RF model. Reducing the number of predictors based on clustering thought is an effective way to reduce the complexity of the RF model, but different dimensionality reduction or feature optimization methods are suitable for different datasets and scenarios. The QPE algorithm established in this paper has high accuracy, so we will not discuss over-fitting in detail in this paper.

At the same time, the over-fitting of machine learning is mainly attributed to the inadequacy of the non-linear model in terms of data extension. Once the training data lacks representativeness, it will have a great impact on the prediction accuracy. Our research data is 48 hours of continuous precipitation, the number of the high precipitation samples in the dataset is relatively low. That is why it is underestimated.

Here we give a possible way to avoid the underestimation of the high precipitation. Based on past precipitation cases, a large number of high precipitation (heavy rain level and above) samples are selected for RF model training, and an algorithm suitable for estimating high precipitation is constructed by correcting model parameters and predictor dimensionality reduction. In actual application, the predictors are input into the universally applicable QPE algorithm established in this paper (which will underestimate high precipitation) and the

high precipitation estimation algorithm respectively. The estimation results of moderate rain level and below in the universally applicable QPE algorithm and the estimation results of heavy rain level and above in the high precipitation estimation algorithm is used as the final precipitation estimation results.

Figure 9: large biases were observed for stations located in mountain areas, maybe the inclusion of DEM as a predictor could account for such biases.

**RESPONSE:** Thanks for your constructive comments. By following your suggestion, we added DEM as a regressor in the model, and large biases over stations located in mountain areas were greatly reduced.